# Explicit Personalization and Local Training: Double Communication Acceleration in Federated Learning

## Abstract

Federated Learning is an evolving machine learning paradigm, in which multiple clients perform computations based on their individual private data, interspersed by communication with a remote server. A common strategy to curtail communication costs is *Local Training*, which consists in performing multiple local stochastic gradient descent steps between successive communication rounds. However, the conventional approach to local training overlooks the practical necessity for client-specific *personalization*, a technique to tailor local models to individual needs. We introduce Scafflix, a novel algorithm that efficiently integrates explicit personalization with local training. This innovative approach benefits from these two techniques, thereby achieving doubly accelerated communication, as we demonstrate both in theory and practice.

## 1 Introduction

Due to privacy concerns and limited computing resources on edge devices, centralized training with all data first gathered in a datacenter is often impossible in many real-world applications. So, Federated Learning (FL) has gained increasing interest as a framework that enables multiple clients to do local computations, based on their personal data kept private, and to communicate back and forth with a server. FL is classically formulated as an empirical risk minimization problem of the form

$$\min_{x \in \mathbb{R}^d} \left[ f(x) := \frac{1}{n} \sum_{i=1}^{n} f_i(x) \right], \qquad \text{(ERM)}$$

where $f_i$ is the local objective on client $i$, $n$ is the total number of clients, $x$ is the global model.

Thus, the usual approach is to solve equation ERM and then to deploy the obtained globally optimal model $x^\star := \arg\min_{x \in \mathbb{R}^d} f(x)$ to all clients. To reduce communication costs between the server and the clients, the practice of updating the local parameters multiple times before aggregation, known as *Local Training* (LT) (Povey et al., 2014; Moritz et al., 2016; McMahan et al., 2017; Li et al., 2020b; Haddadpour & Mahdavi, 2019; Khaled et al., 2019; 2020; Karimireddy et al., 2020; Gorbunov et al., 2020a; Mitra et al., 2021), is widely used in FL. LT, in its most modern form, is a communication-acceleration mechanism, as we detail in Section 2.1.

Meanwhile, there is a growing interest in providing *personalization* to the clients, by providing them more-or-less customized models tailored to their individual needs and heterogeneous data, instead of the one-size-fits-all model $x^\star$. We review existing approaches to personalization in Section 2.2. If personalization is pushed to the extreme, every client just uses its private data to learn its own locally-optimal model

$$x_i^\star := \arg\min_{x \in \mathbb{R}^d} f_i(x)$$

and no communication at all is needed. Thus, intuitively, more personalization means less communication needed to reach a given accuracy. In other words, personalization is a communication-acceleration mechanism, like LT.

Therefore, we raise the following question: *Is it possible to achieve double communication acceleration in FL by jointly leveraging the acceleration potential of personalization and local training?*

For this purpose, we first have to formulate personalized FL as an optimization problem. A compelling interpretation of LT (Hanzely & Richtárik, 2020) is that it amounts to solve an implicit personalization objective of the form:

$$\min_{x_1,\dots,x_n\in\mathbb{R}^d} \frac{1}{n}\sum_{i=1}^n f_i(x_i) + \frac{\lambda}{2n}\sum_{i=1}^n \|\bar{x}-x_i\|^2, \tag{1}$$

where $x_i \in \mathbb{R}^d$ denotes the local model at client $i \in [n] := \{1,\dots,n\}$, $\bar{x} := \frac{1}{n}\sum_{i=1}^n x_i$ is the average of these local models, and $\lambda \geq 0$ is the implicit personalization parameter that controls the amount of personalization. When $\lambda$ is small, the local models tend to be trained locally. On the other hand, a larger $\lambda$ puts more penalty on making the local models $x_i$ close to their mean $\bar{x}$, or equivalently in making all models close to each other, by pushing towards averaging over all clients. Thus, LT is not only compatible with personalization, but can be actually used to implement it, though implicitly: there is a unique parameter $\lambda$ in equation 1 and it is difficult evaluate the amount of personalization for a given value of $\lambda$.

The more accurate FLIX model for personalized FL was proposed by Gasanov et al. (2022). It consists for every client $i$ to first compute locally its personally-optimal model $x_i^\star$,[1] and then to solve the problem

$$\min_{x\in\mathbb{R}^d} \tilde{f}(x) := \frac{1}{n}\sum_{i=1}^n f_i\big(\alpha_i x + (1-\alpha_i)x_i^\star\big), \tag{FLIX}$$

where $\alpha_i \in [0,1]$ is the explicit and individual personalization factor for client $i$. At the end, the personalized model used by client $i$ is the explicit mixture

$$\tilde{x}_i^\star := \alpha_i x^\star + (1-\alpha_i)x_i^\star,$$

where $x^\star$ is the solution to equation FLIX. A smaller value of $\alpha_i$ gives more weight to $x_i^\star$, which means more personalization. On the other hand, if $\alpha_i = 1$, the client $i$ uses the global model $x^\star$ without personalization. Thus, if all $\alpha_i$ are equal to 1, there is no personalization at all and equation FLIX reverts to equation ERM. So, equation FLIX is a more general formulation of FL than equation ERM. The functions in equation FLIX inherit smoothness and strong convexity from the $f_i$, so every algorithm appropriate for equation ERM can also be applied to solve equation FLIX. Gasanov et al. (2022) proposed an algorithm also called FLIX to solve equation FLIX, which is simply vanilla distributed gradient descent (GD) applied to equation FLIX.

In this paper, we first redesign and generalize the recent Scaffnew algorithm (Mishchenko et al., 2022), which features LT and has an accelerated communication complexity, and propose Individualized-Scaffnew (i-Scaffnew), wherein the clients can have different properties. We then apply and tune i-Scaffnew for the problem equation FLIX and propose our new algorithm for personalized FL, which we call Scafflix. We answer positively to the question above and prove that Scafflix enjoys a doubly accelerated communication complexity, by jointly harnessing the acceleration potential of LT and personalization. That is, its communication complexity depends on the square root of the condition number of the functions $f_i$ and on the $\alpha_i$. In addition to establishing the new state of the art for personalized FL with our theoretical guarantees, we show by extensive experiments that Scafflix is efficient in real-world learning setups and outperforms existing algorithms.

Our approach is novel and its good performance is built on a solid theoretical foundation. We stress that our convergence theorem for Scafflix holds under standard assumptions, without bounded variance or any other restriction. By way of comparison with recent works, pFedGate (Chen et al., 2023) bases its theorem on the bounded diversity assumption, which is often unrealistic for non-iid FL. Neither FedCR (Zhang et al., 2023) nor FedGMM (Wu et al., 2023) comes with a conventional convergence theory. pFedGraph (Ye et al., 2023) and FED-PUB (Baek et al., 2023) also lack a robust convergence analysis.

---

[1]In FL, the challenge is to minimize the communication burden, and the running time of local computations, even if not negligible, is very small in comparison. Thus, computing the local optimum to a reasonable accuracy for every client is a completely realistic assumption. In fact, this is nothing but *pretraining*, a common and widely accepted practice in areas like computer vision and NLP. For instance, for the Shakespeare dataset with over 16,000 samples distributed across 1,129 devices, only 50 epochs of local training are needed for every client to attain optimality, as demonstrated in Figure 2. This is noteworthy when comparing with existing methods that require over 800 communication rounds, each encompassing multiple local updates.

## 2 RELATED WORK

### 2.1 LOCAL TRAINING (LT) METHODS IN FEDERATED LEARNING (FL)

Theoretical evolutions of LT in FL have been long-lasting, spanning five generations from empirical results to accelerated communication complexity. The celebrated FedAvg algorithm proposed by McMahan et al. (2017) showed the feasibility of communication-efficient learning from decentralized data. It belongs to the first generation of LT methods, where the focus was on empirical results and practical validations (Povey et al., 2014; Moritz et al., 2016; McMahan et al., 2017).

The second generation of studies on LT for solving equation ERM was based on homogeneity assumptions, such as bounded gradients $\left(\exists c < +\infty, \|\nabla f_i(x)\| \le c, x \in \mathbb{R}^d, i \in [n]\right)$ (Li et al., 2020b) and bounded gradient diversity $\left(\frac{1}{n}\sum_{i=1}^n \|\nabla f_i(x)\|^2 \le c\|\nabla f(x)\|^2\right)$ (Haddadpour & Mahdavi, 2019). However, these assumptions are too restrictive and do not hold in practical FL settings (Kairouz et al., 2019; Wang et al., 2021).

The third generation of approaches, under generic assumptions on the convexity and smoothness of the functions, exhibited sublinear convergence (Khaled et al., 2019; 2020) or linear convergence to a neighborhood (Malinovsky et al., 2020).

Recently, popular algorithms have emerged, such as Scaffold (Karimireddy et al., 2020), S-Local-GD (Gorbunov et al., 2020a), and FedLin (Mitra et al., 2021), successfully correcting for the client drift and enjoying linear convergence to an exact solution under standard assumptions. However, their communication complexity remains the same as with GD, namely $\mathcal{O}(\kappa \log \epsilon^{-1})$, where $\kappa := L/\mu$ is the condition number.

Finally, Scaffnew was proposed by Mishchenko et al. (2022), with accelerated communication complexity $\mathcal{O}(\sqrt{\kappa}\log \epsilon^{-1})$. This is a major achievement, which proves for the first time that LT is a communication acceleration mechanism. Thus, Scaffnew is the first algorithm in what can be considered the fifth generation of LT-based methods with accelerated convergence. Subsequent works have further extended Scaffnew with features such as variance-reduced stochastic gradients (Malinovsky et al., 2022), compression (Condat et al., 2022), partial client participation (Condat et al., 2023), asynchronous communication of different clients (Maranjyan et al., 2022), and to a general primal–dual framework (Condat & Richtárik, 2023). The fifth generation of LT-based methods also includes the 5GCS algorithm (Grudzień et al., 2023), based on a different approach: the local steps correspond to an inner loop to compute a proximity operator inexactly. Our proposed algorithm Scafflix generalizes Scaffnew and enjoys even better accelerated communication complexity, thanks to a better dependence on the possibly different condition numbers of the functions $f_i$.

### 2.2 PERSONALIZATION IN FL

We can distinguish three main approaches to achieve personalization:

a) One-stage training of a single global model using personalization algorithms. One common scheme is to design a suitable regularizer to balance between current and past local models (Li et al., 2021) or between global and local models (Li et al., 2020a; Hanzely & Richtárik, 2020). The FLIX model (Gasanov et al., 2022) achieves explicit personalization by balancing the local and global model using interpolation. Meta-learning is also popular in this area, as evidenced by Dinh et al. (2020), who proposed a federated meta-learning framework using Moreau envelopes and a regularizer to balance personalization and generalization.

b) Training a global model and fine-tuning every local client or knowledge transfer/distillation. This approach allows knowledge transfer from a source domain trained in the FL manner to target domains (Li & Wang, 2019), which is especially useful for personalization in healthcare domains (Chen et al., 2020; Yang et al., 2020).

c) Collaborative training between the global model and local models. The basic idea behind this approach is that each local client trains some personalized parts of a large model, such as the last few layers of a neural network. Parameter decoupling enables learning of task-specific representations for better personalization (Arivazhagan et al., 2019; Bui et al., 2019), while channel sparsity encourages each local client to train the neural network with sparsity based on their limited computation resources (Horvath et al., 2021; Alam et al., 2022; Mei et al., 2022).

---

**Algorithm 1** Scafflix for equation FLIX

1: **input:** stepsizes $\gamma_1 > 0, \ldots, \gamma_n > 0$; probability $p \in (0,1]$; initial estimates $x_1^0, \ldots, x_n^0 \in \mathbb{R}^d$ and $h_1^0, \ldots, h_n^0 \in \mathbb{R}^d$ such that $\sum_{i=1}^n h_i^0 = 0$, personalization weights $\alpha_1, \ldots, \alpha_n$
2: at the server, $\gamma := \left( \frac{1}{n} \sum_{i=1}^n \alpha_i^2 \gamma_i^{-1} \right)^{-1}$          $\diamond$ $\gamma$ is used by the server at Step 11
3: at clients in parallel, $x_i^\star := \arg\min f_i$                 $\diamond$ not needed if $\alpha_i = 1$
4: **for** $t = 0, 1, \ldots$ **do**
5:      flip a coin $\theta^t := \{1$ with probability $p$, $0$ otherwise$\}$
6:      **for** $i = 1, \ldots, n$, at clients in parallel, **do**
7:          $\tilde{x}_i^t := \alpha_i x_i^t + (1 - \alpha_i) x_i^\star$          $\diamond$ estimate of the personalized model $\tilde{x}_i^\star$
8:          compute an estimate $g_i^t$ of $\nabla f_i(\tilde{x}_i^t)$
9:          $\hat{x}_i^t := x_i^t - \frac{\gamma_i}{\alpha_i}\left(g_i^t - h_i^t\right)$                $\diamond$ local SGD step
10:          **if** $\theta^t = 1$ **then**
11:             send $\frac{\alpha_i^2}{\gamma_i} \hat{x}_i^t$ to the server, which aggregates $\bar{x}^t := \frac{\gamma}{n} \sum_{j=1}^n \frac{\alpha_j^2}{\gamma_j} \hat{x}_j^t$ and broadcasts it to all clients          $\diamond$ communication, but only with small probability $p$
12:             $x_i^{t+1} := \bar{x}^t$
13:             $h_i^{t+1} := h_i^t + \frac{p\alpha_i}{\gamma_i}\left(\bar{x}^t - \hat{x}_i^t\right)$          $\diamond$ update of the local control variate $h_i^t$
14:          **else**
15:             $x_i^{t+1} := \hat{x}_i^t$
16:             $h_i^{t+1} := h_i^t$
17:          **end if**
18:      **end for**
19: **end for**

---

Despite the significant progress made in FL personalization, many approaches only present empirical results. Our approach benefits from the simplicity and efficiency of the FLIX framework and enjoys accelerated convergence.

## 3 PROPOSED ALGORITHM Scafflix AND CONVERGENCE ANALYSIS

We generalize Scaffnew (Mishchenko et al., 2022) and propose Individualized-Scaffnew (i-Scaffnew), shown as Algorithm 2 in the Appendix. Its novelty with respect to Scaffnew is to make use of different stepsizes $\gamma_i$ for the local SGD steps, in order to exploit the possibly different values of $L_i$ and $\mu_i$, as well as the different properties $A_i$ and $C_i$ of the stochastic gradients. This change is not straightforward and requires to rederive the whole proof with a different Lyapunov function and to formally endow $\mathbb{R}^d$ with a different inner product at every client.

We then apply and tune i-Scaffnew for the problem equation FLIX and propose our new algorithm for personalized FL, which we call Scafflix, shown as Algorithm 1.

We analyze Scafflix in the strongly convex case, because the analysis of linear convergence rates in this setting gives clear insights and allows us to deepen our theoretical understanding of LT and personalization. And to the best of our knowledge, there is no analysis of Scaffnew in the nonconvex setting. But we conduct several nonconvex deep learning experiments to show that our theoretical findings also hold in practice.

**Assumption 1** (Smoothness and strong convexity). *In the problem equation FLIX (and equation ERM as the particular case $\alpha_i \equiv 1$), we assume that for every $i \in [n]$, the function $f_i$ is $L_i$-smooth and $\mu_i$-strongly convex,[2] for some $L_i \geq \mu_i > 0$. This implies that the problem is strongly convex, so that its solution $x^\star$ exists and is unique.*

We also make the two following assumptions on the stochastic gradients $g_i^t$ used in Scafflix (and i-Scaffnew as a particular case with $\alpha_i \equiv 1$).

---

[2]A function $f : \mathbb{R}^d \to \mathbb{R}$ is said to be $L$-smooth if it is differentiable and its gradient is Lipschitz continuous with constant $L$; that is, for every $x \in \mathbb{R}^d$ and $y \in \mathbb{R}^d$, $\|\nabla f(x) - \nabla f(y)\| \leq L\|x - y\|$, where, here and throughout the paper, the norm is the Euclidean norm. $f$ is said to be $\mu$-strongly convex if $f - \frac{\mu}{2}\|\cdot\|^2$ is convex. We refer to Bauschke & Combettes (2017) for such standard notions of convex analysis.

**Assumption 2** (Unbiasedness). *We assume that for every $t \geq 0$ and $i \in [n]$, $g_i^t$ is an unbiased estimate of $\nabla f_i(\tilde{x}_i^t)$; that is,*

$$\mathbb{E}\big[g_i^t \mid \tilde{x}_i^t\big] = \nabla f_i(\tilde{x}_i^t).$$

To characterize unbiased stochastic gradient estimates, the modern notion of *expected smoothness* is well suited (Gower et al., 2019; Gorbunov et al., 2020b):

**Assumption 3** (Expected smoothness). *We assume that, for every $i \in [n]$, there exist constants $A_i \geq L_i$ [3] and $C_i \geq 0$ such that, for every $t \geq 0$,*

$$\mathbb{E}\Big[\big\|g_i^t - \nabla f_i(\tilde{x}_i^\star)\big\|^2 \mid \tilde{x}_i^t\Big] \leq 2A_i D_{f_i}(\tilde{x}_i^t, \tilde{x}_i^\star) + C_i, \tag{2}$$

*where $D_\varphi(x, x') := f(x) - f(x') - \langle \nabla f(x'), x - x' \rangle \geq 0$ denotes the Bregman divergence of a function $\varphi$ at points $x, x' \in \mathbb{R}^d$.*

Thus, unlike the analysis in Mishchenko et al. (2022, Assumption 4.1), where the same constants are assumed for all clients, since we consider personalization, we individualize the analysis: we consider that each client can be different and use stochastic gradients characterized by its own constants $A_i$ and $C_i$. This is more representative of practical settings. Assumption 3 is general and covers in particular the following two important cases (Gower et al., 2019):

1. (bounded variance) If $g_i^t$ is equal to $\nabla f_i(\tilde{x}_i^t)$ plus a zero-mean random error of variance $\sigma_i^2$ (this covers the case of the exact gradient $g_i^t = \nabla f_i(\tilde{x}_i^t)$ with $\sigma_i = 0$), then Assumption 3 is satisfied with $A_i = L_i$ and $C_i = \sigma_i^2$.

2. (sampling) If $f_i = \frac{1}{n_i}\sum_{j=1}^{n_i} f_{i,j}$ for some $L_i$-smooth functions $f_{i,j}$ and $g_i^t = \nabla f_{i,j^t}(\tilde{x}_i^t)$ for some $j^t$ chosen uniformly at random in $[n_i]$, then Assumption 3 is satisfied with $A_i = 2L_i$ and $C_i = \big(\frac{2}{n_i}\sum_{j=1}^{n_i}\|\nabla f_{i,j}(\tilde{x}_i^\star)\|^2\big) - 2\|\nabla f_i(\tilde{x}_i^\star)\|^2$ (this can be extended to minibatch and nonuniform sampling).

We now present our main convergence result:

**Theorem 1** (fast linear convergence). *In equation FLIX and Scafflix, suppose that Assumptions 1, 2, 3 hold and that for every $i \in [n]$, $0 < \gamma_i \leq \frac{1}{A_i}$. For every $t \geq 0$, define the Lyapunov function*

$$\Psi^t := \frac{1}{n}\sum_{i=1}^{n}\frac{\gamma_{\min}}{\gamma_i}\big\|\tilde{x}_i^t - \tilde{x}_i^\star\big\|^2 + \frac{\gamma_{\min}}{p^2}\frac{1}{n}\sum_{i=1}^{n}\gamma_i\big\|h_i^t - \nabla f_i(\tilde{x}_i^\star)\big\|^2, \tag{3}$$

*where $\gamma_{\min} := \min_{i \in [n]}\gamma_i$. Then Scafflix converges linearly: for every $t \geq 0$,*

$$\mathbb{E}\big[\Psi^t\big] \leq (1-\zeta)^t\Psi^0 + \frac{\gamma_{\min}}{\zeta}\frac{1}{n}\sum_{i=1}^{n}\gamma_i C_i, \tag{4}$$

*where*

$$\zeta = \min\left(\min_{i \in [n]}\gamma_i\mu_i, p^2\right). \tag{5}$$

It is important to note that the range of the stepsizes $\gamma_i$, the Lyapunov function $\Psi^t$ and the convergence rate in equation 4–equation 5 do not depend on the personalization weights $\alpha_i$; they only play a role in the definition of the personalized models $\tilde{x}_i^t$ and $\tilde{x}_i^\star$. Indeed, the convergence speed essentially depends on the conditioning of the functions $x \mapsto f_i(\alpha_i x + (1-\alpha_i)x_i^\star)$, which are independent from the $\alpha_i$. More precisely, let us define, for every $i \in [n]$,

$$\kappa_i := \frac{L_i}{\mu_i} \geq 1 \quad \text{and} \quad \kappa_{\max} = \max_{i \in [n]}\kappa_i,$$

---

[3]We can suppose $A_i \geq L_i$. Indeed, we have the bias-variance decomposition $\mathbb{E}\Big[\big\|g_i^t - \nabla f_i(\tilde{x}_i^t)\big\|^2 \mid \tilde{x}_i^t\Big] = \big\|\nabla f_i(\tilde{x}_i^t) - \nabla f_i(\tilde{x}_i^\star)\big\|^2 + \mathbb{E}\Big[\big\|g_i^t - \nabla f_i(\tilde{x}_i^t)\big\|^2 \mid \tilde{x}_i^t\Big] \geq \big\|\nabla f_i(\tilde{x}_i^t) - \nabla f_i(\tilde{x}_i^\star)\big\|^2$. Assuming that $L_i$ is the best known smoothness constant of $f_i$, we cannot improve the constant $L_i$ such that for every $x \in \mathbb{R}^d$, $\|\nabla f_i(x) - \nabla f_i(\tilde{x}_i^\star)\|^2 \leq 2L_i D_{f_i}(x, \tilde{x}_i^\star)$. Therefore, $A_i$ in equation 2 has to be $\geq L_i$.

and let us study the complexity of of Scafflix to reach $\epsilon$-accuracy, i.e. $\mathbb{E}[\Psi^t] \leq \epsilon$. If, for every $i \in [n]$, $C_i = 0$, $A_i = \Theta(L_i)$, and $\gamma_i = \Theta(\frac{1}{A_i}) = \Theta(\frac{1}{L_i})$, the iteration complexity of Scafflix is

$$\mathcal{O}\left(\left(\kappa_{\max} + \frac{1}{p^2}\right) \log(\Psi^0 \epsilon^{-1})\right). \tag{6}$$

And since communication occurs with probability $p$, the communication complexity of Scafflix is

$$\mathcal{O}\left(\left(p\kappa_{\max} + \frac{1}{p}\right) \log(\Psi^0 \epsilon^{-1})\right). \tag{7}$$

Note that $\kappa_{\max}$ can be much smaller than $\kappa_{\text{global}} := \frac{\max_i L_i}{\min_i \mu_i}$, which is the condition number that appears in the rate of Scaffnew with $\gamma = \frac{1}{\max_i A_i}$. Thus, Scafflix is much more versatile and adapted to FL with heterogeneous data than Scaffnew.

**Corollary 1** (case $C_i \equiv 0$). *In the conditions of Theorem 1, if $p = \Theta\left(\frac{1}{\sqrt{\kappa_{\max}}}\right)$ and, for every $i \in [n]$, $C_i = 0$, $A_i = \Theta(L_i)$, and $\gamma_i = \Theta(\frac{1}{A_i}) = \Theta(\frac{1}{L_i})$, the communication complexity of Scafflix is*

$$\mathcal{O}\left(\sqrt{\kappa_{\max}} \log(\Psi^0 \epsilon^{-1})\right). \tag{8}$$

**Corollary 2** (general stochastic gradients). *In the conditions of Theorem 1, if $p = \sqrt{\min_{i \in [n]} \gamma_i \mu_i}$ and, for every $i \in [n]$,*

$$\gamma_i = \min\left(\frac{1}{A_i}, \frac{\epsilon \mu_{\min}}{2C_i}\right) \tag{9}$$

*(or $\gamma_i := \frac{1}{A_i}$ if $C_i = 0$), where $\mu_{\min} := \min_{j \in [n]} \mu_j$, the iteration complexity of Scafflix is*

$$\mathcal{O}\left(\left(\max_{i \in [n]} \max\left(\frac{A_i}{\mu_i}, \frac{C_i}{\epsilon \mu_{\min} \mu_i}\right)\right) \log(\Psi^0 \epsilon^{-1})\right) = \mathcal{O}\left(\max\left(\max_{i \in [n]} \frac{A_i}{\mu_i}, \max_{i \in [n]} \frac{C_i}{\epsilon \mu_{\min} \mu_i}\right) \log(\Psi^0 \epsilon^{-1})\right) \tag{10}$$

*and its communication complexity is*

$$\mathcal{O}\left(\max\left(\max_{i \in [n]} \sqrt{\frac{A_i}{\mu_i}}, \max_{i \in [n]} \sqrt{\frac{C_i}{\epsilon \mu_{\min} \mu_i}}\right) \log(\Psi^0 \epsilon^{-1})\right). \tag{11}$$

If $A_i = \Theta(L_i)$ uniformly, we have $\max_{i \in [n]} \sqrt{\frac{A_i}{\mu_i}} = \Theta(\sqrt{\kappa_{\max}})$. Thus, we see that thanks to LT, the communication complexity of Scafflix is accelerated, as it depends on $\sqrt{\kappa_{\max}}$ and $\frac{1}{\sqrt{\epsilon}}$.

In the expressions above, the acceleration effect of personalization is not visible: it is "hidden" in $\Psi^0$, because every client computes $x_i^t$ but what matters is its personalized model $\tilde{x}_i^t$, and $\|\tilde{x}_i^t - \tilde{x}_i^\star\|^2 = \alpha_i^2 \|x_i^t - x^\star\|^2$. In particular, assuming that $x_1^0 = \cdots = x_n^0 = x^0$ and $h_i^0 = \nabla f_i(\tilde{x}_i^0)$, we have

$$\Psi^0 \leq \frac{\gamma_{\min}}{n} \|x^0 - x^\star\|^2 \sum_{i=1}^n \alpha_i^2 \left(\frac{1}{\gamma_i} + \frac{\gamma_i L_i^2}{p^2}\right) \leq \left(\max_i \alpha_i^2\right) \frac{\gamma_{\min}}{n} \|x^0 - x^\star\|^2 \sum_{i=1}^n \left(\frac{1}{\gamma_i} + \frac{\gamma_i L_i^2}{p^2}\right),$$

and we see that the contribution of every client to the initial gap $\Psi^0$ is weighted by $\alpha_i^2$. Thus, the smaller the $\alpha_i$, the smaller $\Psi^0$ and the faster the convergence. This is why personalization is an acceleration mechanism in our setting.

## 4 EXPERIMENTS

We first consider a convex logistic regression problem to show that the empirical behavior of Scafflix is in accordance with the theoretical convergence guarantees available in the convex case. Then, we make extensive experiments of training neural networks on large-scale distributed datasets.

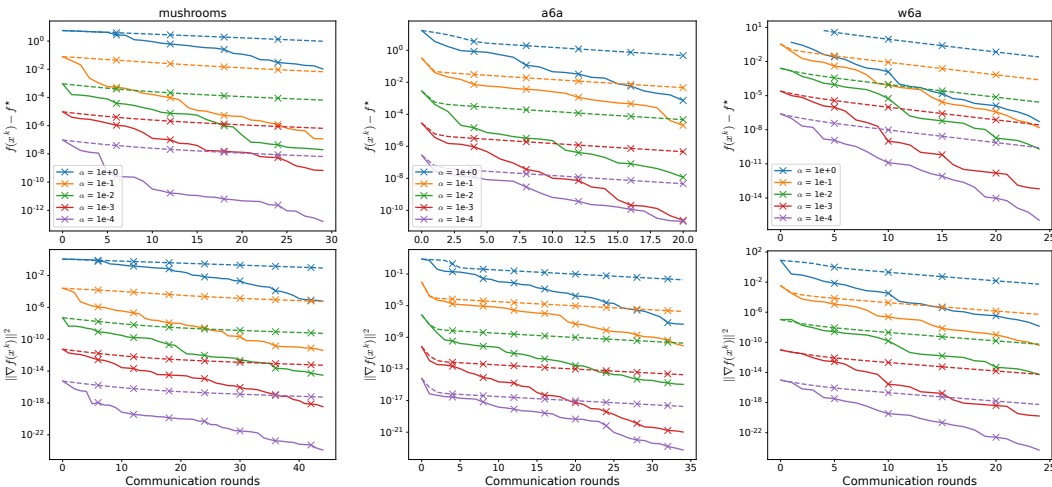

Figure 1: The objective gap $f(x^k) - f^\star$ and the squared gradient norm $\left\|\nabla f(x^k)\right\|^2$ against the number $k$ of communication rounds for Scafflix and GD on the problem equation FLIX. We set all $\alpha_i$ to the same value for simplicity. The dashed line represents GD, while the solid line represents Scafflix. We observe the double communication acceleration achieved through explicit personalization and local training. Specifically, (a) for a given algorithm, smaller $\alpha_i$s (i.e. more personalized models) lead to faster convergence; (b) comparing the two algorithms, Scafflix is faster than GD, thanks to its local training mechanism.

### 4.1 PRELUDE: CONVEX LOGISTIC REGRESSION

We begin our evaluation by considering the standard convex logistic regression problem with an $l_2$ regularizer. This benchmark problem is takes the form equation ERM with

$$f_i(x) := \frac{1}{n_i} \sum_{j=1}^{n_i} \log\left(1 + \exp(-b_{i,j} x^T a_{i,j})\right) + \frac{\mu}{2}\|x\|^2, \tag{12}$$

where $\mu$ represents the regularization parameter, $n_i$ is the total number of data points present at client $i$; $a_{i,j}$ are the training vectors and the $b_{i,j} \in \{-1, 1\}$ are the corresponding labels. Every function $f_i$ is $\mu$-strongly convex and $L_i$-smooth with $L_i = \frac{1}{4n_i}\sum_{j=1}^{n_i}\|a_{i,j}\|^2 + \mu$. We set $\mu$ to 0.1 for this experiment. We employ the `mushrooms`, `a6a`, and `w6a` datasets from the LibSVM library (Chang & Lin, 2011) to conduct these tests. The data is distributed evenly across all clients, and the $\alpha_i$ are set to the same value. The results are shown in Fig. 1. We can observe the double acceleration effect of our approach, which combines explicit personalization and accelerated local training. Lower $\alpha_i$ values, i.e. more personalization, yield faster convergence for both GD and Scafflix. Moreover, Scafflix is much faster than GD, thanks to its specialized local training mechanism.

### 4.2 NEURAL NETWORK TRAINING: DATASETS AND BASELINES FOR EVALUATION

To assess the generalization capabilities of Scafflix, we undertake a comprehensive evaluation involving the training of neural networks using two widely-recognized large-scale FL datasets.

**Datasets.** Our selection comprises two notable large-scale FL datasets: Federated Extended MNIST (FEMNIST) (Caldas et al., 2018), and Shakespeare (McMahan et al., 2017). FEMNIST is a character recognition dataset consisting of 671,585 samples. In accordance with the methodology outlined in FedJax (Ro et al., 2021), we distribute these samples randomly across 3,400 devices. For all algorithms, we employ a Convolutional Neural Network (CNN) model, featuring two convolutional layers and one fully connected layer. The Shakespeare dataset, used for next character prediction tasks, contains a total of 16,068 samples, which we distribute randomly across 1,129 devices. For all algorithms applied to this dataset, we use a Recurrent Neural Network (RNN) model, comprising two Long Short-Term Memory (LSTM) layers and one fully connected layer.

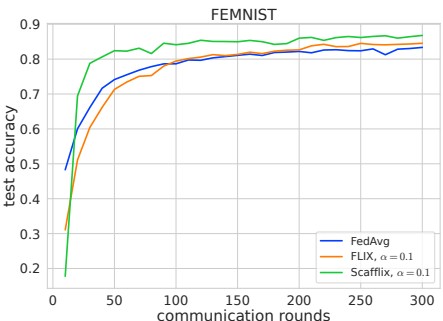 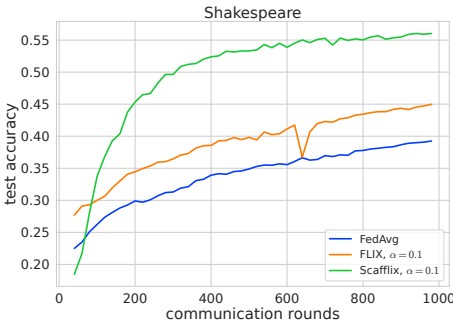

Figure 2: Comparative generalization analysis with baselines. We set the communication probability to $p = 0.2$. The left figure corresponds to the FEMNIST dataset with $\alpha = 0.1$, while the right figure corresponds to the Shakespeare dataset with $\alpha = 0.3$.

**Baselines.** The performance of our proposed Scafflix algorithm is benchmarked against prominent baseline algorithms, specifically FLIX (Gasanov et al., 2022) and FedAvg (McMahan et al., 2016). The FLIX algorithm optimizes the FLIX objective utilizing the SGD method, while FedAvg is designed to optimize the ERM objective. We employ the official implementations for these benchmark algorithms. Comprehensive hyperparameter tuning is carried out for all algorithms, including Scafflix, to ensure optimal results. For both FLIX and Scafflix, local training is required to achieve the local minima for each client. By default, we set the local training batch size at $100$ and employ SGD with a learning rate selected from the set $C_s := \{10^{-5}, 10^{-4}, \cdots, 1\}$. Upon obtaining the local optimum, we execute each algorithm with a batch size of 20 for 1000 communication rounds. The model's learning rate is also selected from the set $C_s$.

## 4.3 ANALYSIS OF GENERALIZATION WITH LIMITED COMMUNICATION ROUNDS

In this section, we perform an in-depth examination of the generalization performance of Scafflix, particularly in scenarios with a limited number of training epochs. This investigation is motivated by our theoretical evidence of the double acceleration property of Scafflix. To that aim, we conduct experiments on both FEMNIST and Shakespeare. These two datasets offer a varied landscape of complexity, allowing for a comprehensive evaluation of our algorithm. In order to ensure a fair comparison with other baseline algorithms, we conducted an extensive search of the optimal hyperparameters for each algorithm. The performance assessment of the generalization capabilities was then carried out on a separate, held-out validation dataset. The hyperparameters that gave the best results in these assessments were selected as the most optimal set.

In order to examine the impact of personalization, we assume that all clients have same $\alpha_i \equiv \alpha$ and we select $\alpha$ in $\{0.1, 0.3, 0.5, 0.7, 0.9\}$. We present the results corresponding to $\alpha = 0.1$ in Fig. 2. Additional comparative analyses with other values of $\alpha$ are available in the Appendix. As shown in Fig. 2, it is clear that Scafflix outperforms the other algorithms in terms of generalization on both the FEMNIST and Shakespeare datasets. Interestingly, the Shakespeare dataset (next-word prediction) poses a greater challenge compared to the FEMNIST dataset (digit recognition). Despite the increased complexity of the task, Scafflix not only delivers significantly better results but also achieves this faster. Thus, Scafflix is superior both in speed and accuracy.

## 4.4 KEY ABLATION STUDIES

In this section, we conduct several critical ablation studies to verify the efficacy of our proposed Scafflix method. These studies investigate the optimal personalization factor for Scafflix, assess the impact of the number of clients per communication round, and examine the influence of the communication probability $p$ in Scafflix.

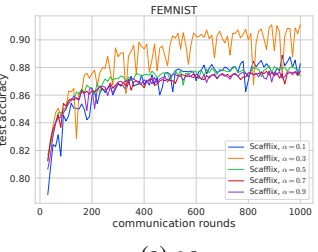 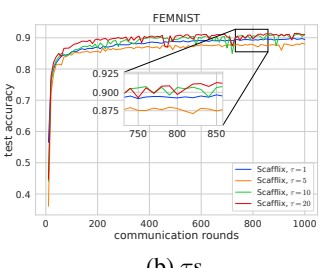 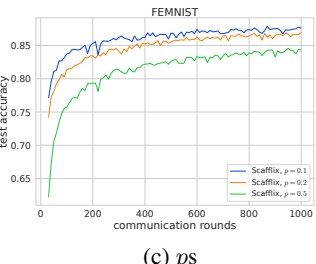

(a) $\alpha$s        (b) $\tau$s        (c) $p$s

Figure 3: Key ablation studies: (a) evaluate the influence of difference personalization factor $\alpha$, (b) examinate the effect of different numbers of clients participating to communication, (c) compare different values of the communication probability $p$.

### 4.4.1 OPTIMAL PERSONALIZATION FACTOR

In this experiment, we explore the effect of varying personalization factors on the FEMNIST dataset. The results are presented in Fig. 3a. We set the batch size to 128 and determine the most suitable learning rate through a hyperparameter search. We consider linearly increasing personalization factors within the set $\{0.1, 0.3, 0.5, 0.7, 0.9\}$. An exponential scale for $\alpha$ is also considered in the Appendix, but the conclusion remains the same.

We note that the optimal personalization factor for the FEMNIST dataset is $0.3$. Interestingly, personalization factors that yield higher accuracy also display a slightly larger variance. However, the overall average performance remains superior. This is consistent with expectations as effective personalization may emphasize the representation of local data, and thus, could be impacted by minor biases in the model parameters received from the server.

### 4.4.2 NUMBER OF CLIENTS COMMUNICATING PER ROUND

In this ablation study, we examine the impact of varying the number of participating clients in each communication round within the Scafflix framework. By default, we set this number to 10. Here, we conduct extensive experiments with different client numbers per round, choosing $\tau$ from $\{1, 5, 10, 20\}$. The results are presented in Fig. 3b. We can observe that Scafflix shows minimal sensitivity to changes in the batch size for local training. However, upon closer examination, we find that larger batch sizes, specifically $\tau = 10$ and $20$, demonstrate slightly improved generalization performance.

### 4.4.3 SELECTION OF COMMUNICATION PROBABILITY $p$

In this ablation study, we explore the effects of varying the communication probability $p$ in Scafflix. We select $p$ from $\{0.1, 0.2, 0.5\}$, and the corresponding results are shown in Fig. 3c. We can clearly see that a smaller value of $p$, indicating reduced communication, facilitates faster convergence and superior generalization performance. This highlights the benefits of LT, which not only makes FL faster and more communication-efficient, but also improves the learning quality.

## 5 CONCLUSION

In the contemporary era of artificial intelligence, improving federated learning to achieve faster convergence and reduce communication costs is crucial to enhance the quality of models trained on huge and heterogeneous datasets. For this purpose, we introduced Scafflix, a novel algorithm that achieves double communication acceleration by redesigning the objective to support explicit personalization for individual clients, while leveraging a state-of-the-art local training mechanism. We provided complexity guarantees in the convex setting, and also validated the effectiveness of our approach in the nonconvex setting through extensive experiments and ablation studies. Thus, our work is a step forward on the important topic of communication-efficient federated learning and offers valuable insights for further investigation in the future.

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

# Appendix

## CONTENTS

---

**Algorithm 2** i-Scaffnew for equation ERM

---

1: **input:** stepsizes $\gamma_1 > 0, \ldots, \gamma_n > 0$; probability $p \in (0, 1]$; initial estimates $x_1^0, \ldots, x_n^0 \in \mathbb{R}^d$ and $h_1^0, \ldots, h_n^0 \in \mathbb{R}^d$ such that $\sum_{i=1}^n h_i^0 = 0$.
2: at the server, $\gamma := \left(\frac{1}{n} \sum_{i=1}^n \gamma_i^{-1}\right)^{-1}$           ⋄ $\gamma$ is used by the server for Step 9
3: **for** $t = 0, 1, \ldots$ **do**
4:     flip a coin $\theta^t := \{1$ with probability $p$, $0$ otherwise$\}$
5:     **for** $i = 1, \ldots, n$, at clients in parallel, **do**
6:        compute an estimate $g_i^t$ of $\nabla f_i(x_i^t)$
7:        $\hat{x}_i^t := x_i^t - \gamma_i \left(g_i^t - h_i^t\right)$                ⋄ local SGD step
8:        **if** $\theta^t = 1$ **then**
9:           send $\frac{1}{\gamma_i} \hat{x}_i^t$ to the server, which aggregates $\bar{x}^t := \frac{\gamma}{n} \sum_{j=1}^n \frac{1}{\gamma_i} \hat{x}_j^t$ and broadcasts it to all clients           ⋄ communication, but only with small probability $p$
10:           $x_i^{t+1} := \bar{x}^t$
11:           $h_i^{t+1} := h_i^t + \frac{p}{\gamma_i}\left(\bar{x}^t - \hat{x}_i^t\right)$         ⋄ update of the local control variate $h_i^t$
12:        **else**
13:           $x_i^{t+1} := \hat{x}_i^t$
14:           $h_i^{t+1} := h_i^t$
15:        **end if**
16:     **end for**
17: **end for**

---

## A    PROPOSED i-Scaffnew ALGORITHM

We consider solving equation ERM with the proposed i-Scaffnew algorithm, shown as Algorithm 2 (applying i-Scaffnew to equation FLIX yields Scafflix, as we discuss subsequently in Section B).

**Theorem 2** (fast linear convergence). *In equation ERM and* i-Scaffnew, *suppose that Assumptions 1, 2, 3 hold and that for every $i \in [n]$, $0 < \gamma_i \leq \frac{1}{A_i}$. For every $t \geq 0$, define the Lyapunov function*

$$\Psi^t := \sum_{i=1}^n \frac{1}{\gamma_i} \left\|x_i^t - x^\star\right\|^2 + \frac{1}{p^2} \sum_{i=1}^n \gamma_i \left\|h_i^t - \nabla f_i(x^\star)\right\|^2. \tag{13}$$

*Then* i-Scaffnew *converges linearly: for every $t \geq 0$,*

$$\mathbb{E}\left[\Psi^t\right] \leq (1 - \zeta)^t \Psi^0 + \frac{1}{\zeta} \sum_{i=1}^n \gamma_i C_i, \tag{14}$$

*where*

$$\zeta = \min\left(\min_{i \in [n]} \gamma_i \mu_i, p^2\right). \tag{15}$$

*Proof.* To simplify the analysis of i-Scaffnew, we introduce vector notations: the problem equation ERM can be written as

$$\text{find } \mathbf{x}^\star = \arg\min_{\mathbf{x} \in \mathcal{X}} \mathbf{f}(\mathbf{x}) \quad \text{s.t.} \quad W\mathbf{x} = 0, \tag{16}$$

where $\mathcal{X} := \mathbb{R}^{d \times n}$, an element $\mathbf{x} = (x_i)_{i=1}^n \in \mathcal{X}$ is a collection of vectors $x_i \in \mathbb{R}^d$, $\mathbf{f} : \mathbf{x} \in \mathcal{X} \mapsto \sum_{i=1}^n f_i(x_i)$, the linear operator $W : \mathcal{X} \to \mathcal{X}$ maps $\mathbf{x} = (x_i)_{i=1}^n$ to $(x_i - \frac{1}{n}\sum_{j=1}^n \frac{\gamma}{\gamma_j} x_j)_{i=1}^n$, for given values $\gamma_1 > 0, \ldots, \gamma_n > 0$ and their harmonic mean $\gamma = \left(\frac{1}{n}\sum_{i=1}^n \gamma_i^{-1}\right)^{-1}$. The constraint $W\mathbf{x} = 0$ means that $\mathbf{x}$ minus its weighted average is zero; that is, $\mathbf{x}$ has identical components $x_1 = \cdots = x_n$. Thus, equation 16 is indeed equivalent to equation ERM. $\mathbf{x}^\star := (x^\star)_{i=1}^n \in \mathcal{X}$ is the unique solution to equation 16, where $x^\star$ is the unique solution to equation ERM.

Moreover, we introduce the weighted inner product in $\mathcal{X}$: $(\mathbf{x}, \mathbf{y}) \mapsto \langle \mathbf{x}, \mathbf{y} \rangle_\gamma := \sum_{i=1}^n \frac{1}{\gamma_i} \langle x_i, y_i \rangle$. Then, the orthogonal projector $P$ onto the hyperspace $\{\mathbf{y} \in \mathcal{X} : y_1 = \cdots = y_n\}$, with respect to this weighted inner product, is $P : \mathbf{x} \in \mathcal{X} \mapsto \bar{\mathbf{x}} = (\bar{x})_{i=1}^n$ with $\bar{x} = \frac{\gamma}{n}\sum_{i=1}^n \frac{1}{\gamma_i} x_i$ (because $\bar{x}$

minimizes $\|\bar{\mathbf{x}} - \mathbf{x}\|_{\gamma}^2$, so that $\frac{1}{n}\sum_{i=1}^n \frac{1}{\gamma_i}(\bar{x} - x_i) = 0$). Thus, $P$, as well as $W = \mathrm{Id} - P$, where $\mathrm{Id}$ denotes the identity, are self-adjoint and positive linear operators with respect to the weighted inner product. Moreover, for every $\mathbf{x} \in \mathcal{X}$,

$$\|\mathbf{x}\|_{\gamma}^2 = \|P\mathbf{x}\|_{\gamma}^2 + \|W\mathbf{x}\|_{\gamma}^2 = \|\bar{\mathbf{x}}\|_{\gamma}^2 + \|W\mathbf{x}\|_{\gamma}^2 = \frac{n}{\gamma}\|\bar{x}\|^2 + \|W\mathbf{x}\|_{\gamma}^2,$$

where $\bar{\mathbf{x}} = (\bar{x})_{i=1}^n$ and $\bar{x} = \frac{\gamma}{n}\sum_{i=1}^n \frac{1}{\gamma_i}x_i$.

Let us introduce further vector notations for the variables of i-Scaffnew: for every $t \geq 0$, we define the *scaled* concatenated control variate $\mathbf{h}^t := (\gamma_i h_i^t)_{i=1}^n$, $\mathbf{h}^\star := (\gamma_i h_i^\star)_{i=1}^n$, with $h_i^\star := \nabla f_i(x^\star)$, $\bar{\mathbf{x}}^t := (\bar{x}^t)_{i=1}^n$, $\mathbf{w}^t := (w_i^t)_{i=1}^n$, with $w_i^t := x_i^t - \gamma_i g_i^t$, $\mathbf{w}^\star := (w_i^\star)_{i=1}^n$, with $w_i^\star := x_i^\star - \gamma_i \nabla f_i(x_i^\star)$, $\hat{\mathbf{h}}^t := \mathbf{h}^t - pW\hat{\mathbf{x}}^t$. Finally, we denote by $\mathcal{F}_0^t$ the $\sigma$-algebra generated by the collection of $\mathcal{X}$-valued random variables $\mathbf{x}^0, \mathbf{h}^0, \ldots, \mathbf{x}^t, \mathbf{h}^t$ and by $\mathcal{F}^t$ the $\sigma$-algebra generated by these variables, as well as the stochastic gradients $g_i^t$.

We can then rewrite the iteration of i-Scaffnew as:

$\quad \hat{\mathbf{x}}^t := \mathbf{w}^t + \mathbf{h}^t$
$\quad$**if** $\theta^t = 1$ **then**
$\quad\quad \mathbf{x}^{t+1} := \bar{\mathbf{x}}^t$
$\quad\quad \mathbf{h}^{t+1} := \mathbf{h}^t - pW\hat{\mathbf{x}}^t$
$\quad$**else**
$\quad\quad \mathbf{x}^{t+1} := \hat{\mathbf{x}}^t$
$\quad\quad \mathbf{h}^{t+1} := \mathbf{h}^t$
$\quad$**end if**

We suppose that $\sum_{i=1}^n h_i^0 = 0$. Then, it follows from the definition of $\bar{x}^t$ that $\frac{\gamma}{n}\sum_{j=1}^n \frac{1}{\gamma_i}(\bar{x}^t - \hat{x}_j^t) = 0$, so that for every $t \geq 0$, $\sum_{i=1}^n h_i^t = 0$; that is, $W\mathbf{h}^t = \mathbf{h}^t$.

Let $t \geq 0$. We have

$$\mathbb{E}\left[\|\mathbf{x}^{t+1} - \mathbf{x}^\star\|_{\gamma}^2 \mid \mathcal{F}^t\right] = p\|\bar{\mathbf{x}}^t - \mathbf{x}^\star\|_{\gamma}^2 + (1-p)\|\hat{\mathbf{x}}^t - \mathbf{x}^\star\|_{\gamma}^2,$$

with

$$\|\bar{\mathbf{x}}^t - \mathbf{x}^\star\|_{\gamma}^2 = \|\hat{\mathbf{x}}^t - \mathbf{x}^\star\|_{\gamma}^2 - \|W\hat{\mathbf{x}}^t\|_{\gamma}^2.$$

Moreover,

$$
\begin{aligned}
\|\hat{\mathbf{x}}^t - \mathbf{x}^\star\|_{\gamma}^2 &= \|\mathbf{w}^t - \mathbf{w}^\star\|_{\gamma}^2 + \|\mathbf{h}^t - \mathbf{h}^\star\|_{\gamma}^2 + 2\langle\mathbf{w}^t - \mathbf{w}^\star, \mathbf{h}^t - \mathbf{h}^\star\rangle_{\gamma} \\
&= \|\mathbf{w}^t - \mathbf{w}^\star\|_{\gamma}^2 - \|\mathbf{h}^t - \mathbf{h}^\star\|_{\gamma}^2 + 2\langle\hat{\mathbf{x}}^t - \mathbf{x}^\star, \mathbf{h}^t - \mathbf{h}^\star\rangle_{\gamma} \\
&= \|\mathbf{w}^t - \mathbf{w}^\star\|_{\gamma}^2 - \|\mathbf{h}^t - \mathbf{h}^\star\|_{\gamma}^2 + 2\langle\hat{\mathbf{x}}^t - \mathbf{x}^\star, \hat{\mathbf{h}}^t - \mathbf{h}^\star\rangle_{\gamma} - 2\langle\hat{\mathbf{x}}^t - \mathbf{x}^\star, \hat{\mathbf{h}}^t - \mathbf{h}^t\rangle_{\gamma} \\
&= \|\mathbf{w}^t - \mathbf{w}^\star\|_{\gamma}^2 - \|\mathbf{h}^t - \mathbf{h}^\star\|_{\gamma}^2 + 2\langle\hat{\mathbf{x}}^t - \mathbf{x}^\star, \hat{\mathbf{h}}^t - \mathbf{h}^\star\rangle_{\gamma} + 2p\langle\hat{\mathbf{x}}^t - \mathbf{x}^\star, W\hat{\mathbf{x}}^t\rangle_{\gamma} \\
&= \|\mathbf{w}^t - \mathbf{w}^\star\|_{\gamma}^2 - \|\mathbf{h}^t - \mathbf{h}^\star\|_{\gamma}^2 + 2\langle\hat{\mathbf{x}}^t - \mathbf{x}^\star, \hat{\mathbf{h}}^t - \mathbf{h}^\star\rangle_{\gamma} + 2p\|W\hat{\mathbf{x}}^t\|_{\gamma}^2.
\end{aligned}
$$

Hence,

$$
\begin{aligned}
\mathbb{E}\left[\|\mathbf{x}^{t+1} - \mathbf{x}^\star\|_{\gamma}^2 \mid \mathcal{F}^t\right] &= \|\hat{\mathbf{x}}^t - \mathbf{x}^\star\|_{\gamma}^2 - p\|W\hat{\mathbf{x}}^t\|_{\gamma}^2 \\
&= \|\mathbf{w}^t - \mathbf{w}^\star\|_{\gamma}^2 - \|\mathbf{h}^t - \mathbf{h}^\star\|_{\gamma}^2 + 2\langle\hat{\mathbf{x}}^t - \mathbf{x}^\star, \hat{\mathbf{h}}^t - \mathbf{h}^\star\rangle_{\gamma} + p\|W\hat{\mathbf{x}}^t\|_{\gamma}^2.
\end{aligned}
$$

On the other hand, we have

$$\mathbb{E}\left[\|\mathbf{h}^{t+1} - \mathbf{h}^\star\|_{\gamma}^2 \mid \mathcal{F}^t\right] = p\|\hat{\mathbf{h}}^t - \mathbf{h}^\star\|_{\gamma}^2 + (1-p)\|\mathbf{h}^t - \mathbf{h}^\star\|_{\gamma}^2$$

and

$$
\begin{aligned}
\left\|\hat{\mathbf{h}}^t - \mathbf{h}^\star\right\|_{\boldsymbol{\gamma}}^2 &= \left\|(\mathbf{h}^t - \mathbf{h}^\star) + (\hat{\mathbf{h}}^t - \mathbf{h}^t)\right\|_{\boldsymbol{\gamma}}^2 \\
&= \left\|\mathbf{h}^t - \mathbf{h}^\star\right\|_{\boldsymbol{\gamma}}^2 + \left\|\hat{\mathbf{h}}^t - \mathbf{h}^t\right\|_{\boldsymbol{\gamma}}^2 + 2\langle \mathbf{h}^t - \mathbf{h}^\star, \hat{\mathbf{h}}^t - \mathbf{h}^t \rangle_{\boldsymbol{\gamma}} \\
&= \left\|\mathbf{h}^t - \mathbf{h}^\star\right\|_{\boldsymbol{\gamma}}^2 - \left\|\hat{\mathbf{h}}^t - \mathbf{h}^t\right\|_{\boldsymbol{\gamma}}^2 + 2\langle \hat{\mathbf{h}}^t - \mathbf{h}^\star, \hat{\mathbf{h}}^t - \mathbf{h}^t \rangle_{\boldsymbol{\gamma}} \\
&= \left\|\mathbf{h}^t - \mathbf{h}^\star\right\|_{\boldsymbol{\gamma}}^2 - \left\|\hat{\mathbf{h}}^t - \mathbf{h}^t\right\|_{\boldsymbol{\gamma}}^2 - 2p\langle \hat{\mathbf{h}}^t - \mathbf{h}^\star, W(\hat{\mathbf{x}}^t - \mathbf{x}^\star) \rangle_{\boldsymbol{\gamma}} \\
&= \left\|\mathbf{h}^t - \mathbf{h}^\star\right\|_{\boldsymbol{\gamma}}^2 - p^2 \left\|W\hat{\mathbf{x}}^t\right\|_{\boldsymbol{\gamma}}^2 - 2p\langle W(\hat{\mathbf{h}}^t - \mathbf{h}^\star), \hat{\mathbf{x}}^t - \mathbf{x}^\star \rangle_{\boldsymbol{\gamma}} \\
&= \left\|\mathbf{h}^t - \mathbf{h}^\star\right\|_{\boldsymbol{\gamma}}^2 - p^2 \left\|W\hat{\mathbf{x}}^t\right\|_{\boldsymbol{\gamma}}^2 - 2p\langle \hat{\mathbf{h}}^t - \mathbf{h}^\star, \hat{\mathbf{x}}^t - \mathbf{x}^\star \rangle_{\boldsymbol{\gamma}}.
\end{aligned}
$$

Hence,

$$
\begin{aligned}
\mathbb{E}&\left[\left\|\mathbf{x}^{t+1} - \mathbf{x}^\star\right\|_{\boldsymbol{\gamma}}^2 \mid \mathcal{F}^t\right] + \frac{1}{p^2}\mathbb{E}\left[\left\|\mathbf{h}^{t+1} - \mathbf{h}^\star\right\|_{\boldsymbol{\gamma}}^2 \mid \mathcal{F}^t\right] \\
&= \left\|\mathbf{w}^t - \mathbf{w}^\star\right\|_{\boldsymbol{\gamma}}^2 - \left\|\mathbf{h}^t - \mathbf{h}^\star\right\|_{\boldsymbol{\gamma}}^2 + 2\langle \hat{\mathbf{x}}^t - \mathbf{x}^\star, \hat{\mathbf{h}}^t - \mathbf{h}^\star \rangle_{\boldsymbol{\gamma}} + p\left\|W\hat{\mathbf{x}}^t\right\|_{\boldsymbol{\gamma}}^2 \\
&\quad + \frac{1}{p^2}\left\|\mathbf{h}^t - \mathbf{h}^\star\right\|_{\boldsymbol{\gamma}}^2 - p\left\|W\hat{\mathbf{x}}^t\right\|_{\boldsymbol{\gamma}}^2 - 2\langle \hat{\mathbf{h}}^t - \mathbf{h}^\star, \hat{\mathbf{x}}^t - \mathbf{x}^\star \rangle_{\boldsymbol{\gamma}} \\
&= \left\|\mathbf{w}^t - \mathbf{w}^\star\right\|_{\boldsymbol{\gamma}}^2 + \frac{1}{p^2}\left(1 - p^2\right)\left\|\mathbf{h}^t - \mathbf{h}^\star\right\|_{\boldsymbol{\gamma}}^2.
\end{aligned}
\tag{17}
$$

Moreover, for every $i \in [n]$,

$$
\begin{aligned}
\left\|w_i^t - w_i^\star\right\|^2 &= \left\|x_i^t - x^\star - \gamma_i\big(g_i^t - \nabla f_i(x^\star)\big)\right\|^2 \\
&= \left\|x_i^t - x^\star\right\|^2 - 2\gamma_i\langle x_i^t - x^\star, g_i^t - \nabla f_i(x^\star)\rangle + \gamma_i^2\left\|g_i^t - \nabla f_i(x^\star)\right\|^2,
\end{aligned}
$$

and, by unbiasedness of $g_i^t$ and Assumption 2,

$$
\begin{aligned}
\mathbb{E}\left[\left\|w_i^t - w_i^\star\right\|^2 \mid \mathcal{F}_0^t\right] &= \left\|x_i^t - x^\star\right\|^2 - 2\gamma_i\langle x_i^t - x^\star, \nabla f_i(x_i^t) - \nabla f_i(x^\star)\rangle \\
&\quad + \gamma_i^2\mathbb{E}\left[\left\|g_i^t - \nabla f_i(x^\star)\right\|^2 \mid \mathcal{F}^t\right] \\
&\leq \left\|x_i^t - x^\star\right\|^2 - 2\gamma_i\langle x_i^t - x^\star, \nabla f_i(x_i^t) - \nabla f_i(x^\star)\rangle + 2\gamma_i^2 A_i D_{f_i}(x_i^t, x^\star) \\
&\quad + \gamma_i^2 C_i.
\end{aligned}
$$

It is easy to see that $\langle x_i^t - x^\star, \nabla f_i(x_i^t) - \nabla f_i(x^\star)\rangle = D_{f_i}(x_i^t, x^\star) + D_{f_i}(x^\star, x_i^t)$. This yields

$$
\begin{aligned}
\mathbb{E}\left[\left\|w_i^t - w_i^\star\right\|^2 \mid \mathcal{F}_0^t\right] &\leq \left\|x_i^t - x^\star\right\|^2 - 2\gamma_i D_{f_i}(x^\star, x_i^t) - 2\gamma_i D_{f_i}(x_i^t, x^\star) + 2\gamma_i^2 A_i D_{f_i}(x_i^t, x^\star) \\
&\quad + \gamma_i^2 C_i.
\end{aligned}
$$

In addition, the strong convexity of $f_i$ implies that $D_{f_i}(x^\star, x_i^t) \geq \frac{\mu_i}{2}\left\|x_i^t - x^\star\right\|^2$, so that

$$
\mathbb{E}\left[\left\|w_i^t - w_i^\star\right\|^2 \mid \mathcal{F}_0^t\right] \leq (1 - \gamma_i\mu_i)\left\|x_i^t - x^\star\right\|^2 - 2\gamma_i(1 - \gamma_i A_i)D_{f_i}(x_i^t, x^\star) + \gamma_i^2 C_i,
$$

and since we have supposed $\gamma_i \leq \frac{1}{A_i}$,

$$
\mathbb{E}\left[\left\|w_i^t - w_i^\star\right\|^2 \mid \mathcal{F}_0^t\right] \leq (1 - \gamma_i\mu_i)\left\|x_i^t - x^\star\right\|^2 + \gamma_i^2 C_i.
$$

Therefore,

$$
\mathbb{E}\left[\left\|\mathbf{w}^t - \mathbf{w}^\star\right\|_{\boldsymbol{\gamma}}^2 \mid \mathcal{F}_0^t\right] \leq \max_{i \in [n]}(1 - \gamma_i\mu_i)\left\|\mathbf{x}^t - \mathbf{x}^\star\right\|_{\boldsymbol{\gamma}}^2 + \sum_{i=1}^n \gamma_i C_i
$$

and

$$\mathbb{E}\big[\Psi^{t+1} \mid \mathcal{F}_0^t\big] = \mathbb{E}\Big[\big\|\mathbf{x}^{t+1} - \mathbf{x}^\star\big\|_{\boldsymbol{\gamma}}^2 \mid \mathcal{F}_0^t\Big] + \frac{1}{p^2}\mathbb{E}\Big[\big\|\mathbf{h}^{t+1} - \mathbf{h}^\star\big\|_{\boldsymbol{\gamma}}^2 \mid \mathcal{F}_0^t\Big]$$

$$\leq \max_{i \in [n]}(1 - \gamma_i\mu_i)\big\|\mathbf{x}^t - \mathbf{x}^\star\big\|_{\boldsymbol{\gamma}}^2 + \frac{1}{p^2}\left(1 - p^2\right)\big\|\mathbf{h}^t - \mathbf{h}^\star\big\|_{\boldsymbol{\gamma}}^2 + \sum_{i=1}^n \gamma_i C_i$$

$$\leq (1 - \zeta)\left(\big\|\mathbf{x}^t - \mathbf{x}^\star\big\|_{\boldsymbol{\gamma}}^2 + \frac{1}{p^2}\big\|\mathbf{h}^t - \mathbf{h}^\star\big\|_{\boldsymbol{\gamma}}^2\right) + \sum_{i=1}^n \gamma_i C_i$$

$$= (1 - \zeta)\Psi^t + \sum_{i=1}^n \gamma_i C_i, \tag{18}$$

where

$$\zeta = \min\left(\min_{i \in [n]} \gamma_i\mu_i, p^2\right).$$

Using the tower rule, we can unroll the recursion in equation 18 to obtain the unconditional expectation of $\Psi^{t+1}$. □

## B    From i-Scaffnew to Scafflix

We suppose that Assumptions 1, 2, 3 hold. We define for every $i \in [n]$ the function $\tilde{f}_i : x \in \mathbb{R}^d \mapsto f_i\big(\alpha_i x + (1 - \alpha_i)x_i^\star\big)$. Thus, equation FLIX takes the form of equation ERM with $f_i$ replaced by $\tilde{f}_i$.

We want to derive Scafflix from i-Scaffnew applied to equation ERM with $f_i$ replaced by $\tilde{f}_i$. For this, we first observe that for every $i \in [n]$, $\tilde{f}_i$ is $\alpha_i^2 L_i$-smooth and $\alpha_i^2 \mu_i$-strongly convex. This follows easily from the fact that $\nabla\tilde{f}_i(x) = \alpha_i \nabla f_i\big(\alpha_i x + (1 - \alpha_i)x_i^\star\big)$.

Second, for every $t \geq 0$ and $i \in [n]$, $g_i^t$ is an unbiased estimate of $\nabla f_i(\tilde{x}_i^t) = \alpha_i^{-1}\nabla\tilde{f}_i(x_i^t)$. Therefore, $\alpha_i g_i^t$ is an unbiased estimate of $\nabla\tilde{f}_i(x_i^t)$ satisfying

$$\mathbb{E}\left[\Big\|\alpha_i g_i^t - \nabla\tilde{f}_i(x^\star)\Big\|^2 \mid x_i^t\right] = \alpha_i^2\mathbb{E}\Big[\big\|g_i^t - \nabla f_i(\tilde{x}_i^\star)\big\|^2 \mid \tilde{x}_i^t\Big] \leq 2\alpha_i^2 A_i D_{f_i}(\tilde{x}_i^t, \tilde{x}_i^\star) + \alpha_i^2 C_i.$$

Moreover,

$$
\begin{aligned}
D_{f_i}(\tilde{x}_i^t, \tilde{x}_i^\star) &= f_i(\tilde{x}_i^t) - f_i(\tilde{x}_i^\star) - \langle\nabla f_i(\tilde{x}_i^\star), \tilde{x}_i^t - \tilde{x}_i^\star\rangle \\
&= \tilde{f}_i(x_i^t) - \tilde{f}_i(x^\star) - \langle\alpha_i^{-1}\nabla\tilde{f}_i(x^\star), \alpha_i(x_i^t - x^\star)\rangle \\
&= \tilde{f}_i(x_i^t) - \tilde{f}_i(x^\star) - \langle\nabla\tilde{f}_i(x^\star), x_i^t - x^\star\rangle \\
&= D_{\tilde{f}_i}(x_i^t, x^\star).
\end{aligned}
$$

Thus, we obtain Scafflix by applying i-Scaffnew to solve equation FLIX, viewed as equation ERM with $f_i$ replaced by $\tilde{f}_i$, and further making the following substitutions in the algorithm: $g_i^t$ is replaced by $\alpha_i g_i^t$, $h_i^t$ is replaced by $\alpha_i h_i^t$ (so that $h_i^t$ in Scafflix converges to $\nabla f_i(\tilde{x}_i^\star)$ instead of $\nabla\tilde{f}_i(x^\star) = \alpha_i\nabla f_i(\tilde{x}_i^\star)$), $\gamma_i$ is replaced by $\alpha_i^{-2}\gamma_i$ (so that the $\alpha_i$ disappear in the theorem).

Accordingly, Theorem 1 follows from Theorem 2, with the same substitutions and with $A_i$, $C_i$ and $\mu_i$ replaced by $\alpha_i^2 A_i$, $\alpha_i^2 C_i$ and $\alpha_i^2\mu_i$, respectively. Finally, the Lyapunov function is multiplied by $\gamma_{\min}/n$ to make it independent from $\epsilon$ when scaling the $\gamma_i$ by $\epsilon$ in Corollary 2.

We note that i-Scaffnew is recovered as a particular case of Scafflix if $\alpha_i \equiv 1$, so that Scafflix is indeed more general.

## C    Proof of Corollary 2

We place ourselves in the conditions of Theorem 1. Let $\epsilon > 0$. We want to choose the $\gamma_i$ and the number of iterations $T \geq 0$ such that $\mathbb{E}\big[\Psi^T\big] \leq \epsilon$. For this, we bound the two terms $(1 - \zeta)^T\Psi^0$ and $\frac{\gamma_{\min}}{\zeta n}\sum_{i=1}^n \gamma_i C_i$ in equation 4 by $\epsilon/2$.

We set $p = \sqrt{\min_{i \in [n]} \gamma_i \mu_i}$, so that $\zeta = \min_{i \in [n]} \gamma_i \mu_i$. We have

$$T \geq \frac{1}{\zeta} \log(2\Psi^0 \epsilon^{-1}) \Rightarrow (1 - \zeta)^T \Psi^0 \leq \frac{\epsilon}{2}. \tag{19}$$

Moreover,

$$(\forall i \in [n] \text{ s.t. } C_i > 0) \; \gamma_i \leq \frac{\epsilon \mu_{\min}}{2 C_i} \Rightarrow \frac{\gamma_{\min}}{\zeta n} \sum_{i=1}^{n} \gamma_i C_i \leq \frac{\epsilon}{2} \frac{\left( \min_{j \in [n]} \gamma_j \right) \left( \min_{j \in [n]} \mu_j \right)}{\min_{j \in [n]} \gamma_j \mu_j} \leq \frac{\epsilon}{2}.$$

Therefore, we set for every $i \in [n]$

$$\gamma_i := \min \left( \frac{1}{A_i}, \frac{\epsilon \mu_{\min}}{2 C_i} \right)$$

(or $\gamma_i := \frac{1}{A_i}$ if $C_i = 0$), and we get from equation 19 that $\mathbb{E}[\Psi^T] \leq \epsilon$ after

$$\mathcal{O} \left( \left( \max_{i \in [n]} \max \left( \frac{A_i}{\mu_i}, \frac{C_i}{\epsilon \mu_{\min} \mu_i} \right) \right) \log(\Psi^0 \epsilon^{-1}) \right)$$

iterations.

## D  ADDITIONAL EXPERIMENTAL RESULTS

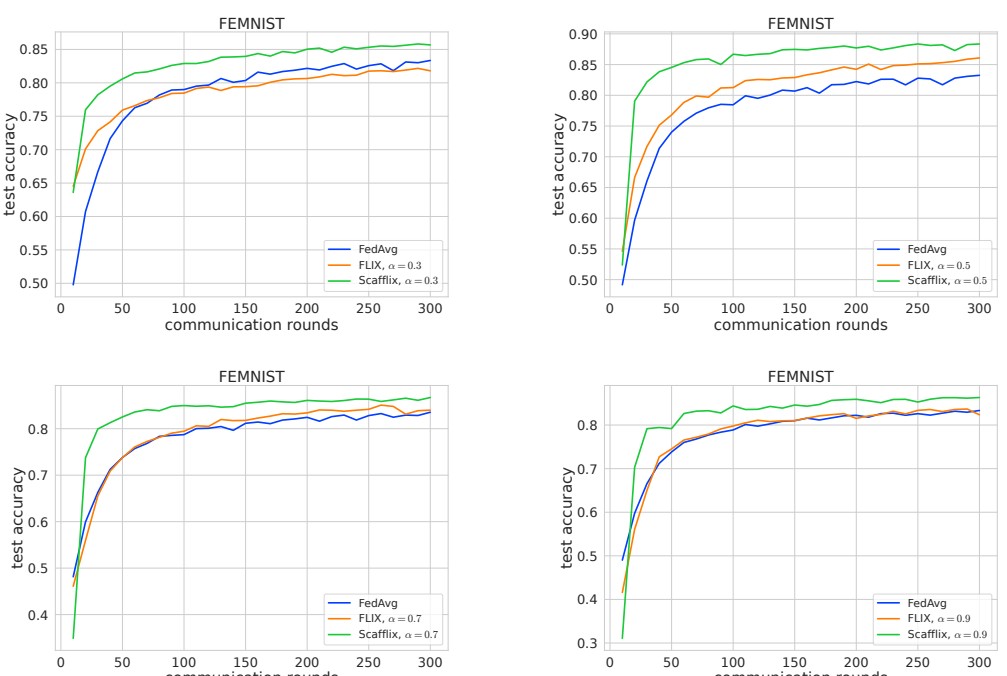

Figure 4: As part of our experimentation on the FEMNIST dataset, we performed complementary ablations by incorporating various personalization factors, represented as $\alpha$. In the main section, we present the results obtained specifically with $\alpha = 0.1$. Furthermore, we extend our analysis by highlighting the outcomes achieved with $\alpha$ values spanning from 0.3 to 0.9, inclusively.

## E  ADDITIONAL BASELINES

While our research primarily seeks to ascertain the impact of explicit personalization and local training on communication costs, we recognize the interest of the community for a broad comparative

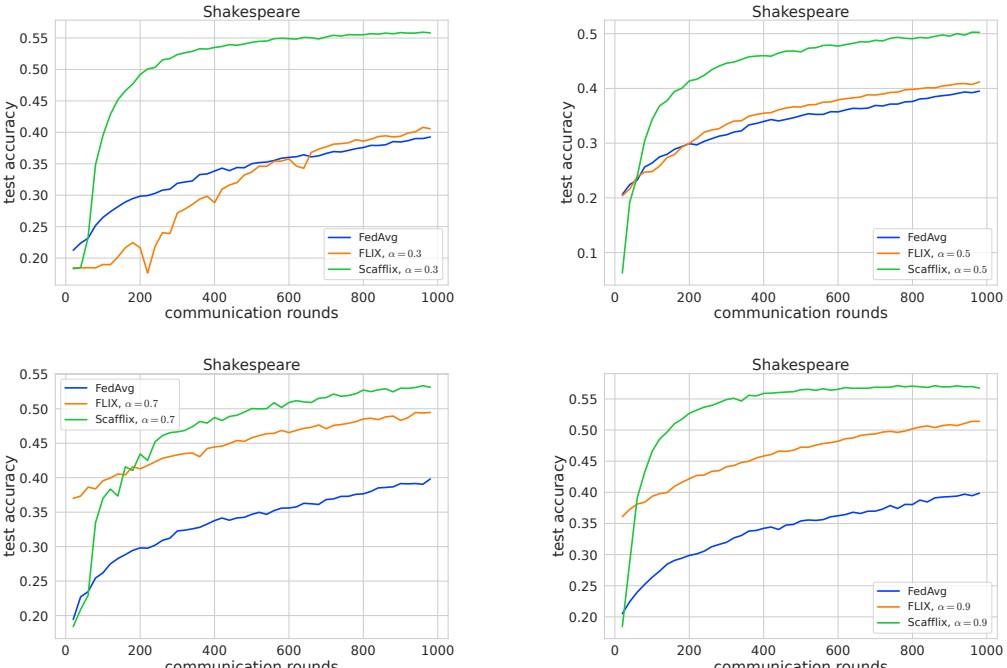

Figure 5: In our investigation of the Shakespeare dataset, we carried out complementary ablations, considering a range of personalization factors denoted as $\alpha$. The selection strategy for determining the appropriate $\alpha$ values remains consistent with the methodology described in the above figure.

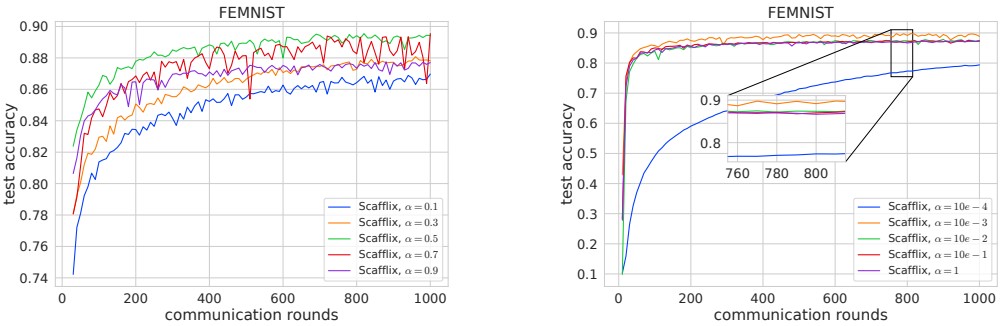

Figure 6: Ablation studies with different values of the personalization factor $\alpha$. The left figure is the complementary experiment of linearly increasing $\alpha$ with full batch size; the right is the figure with exponentially increasing $\alpha$ with default batch size 20.

Table 1: Results of additional baselines.

| Method | Ditto | FedSR-FT | FedPAC | FedCR | Scafflix |
|---|---|---|---|---|---|
| CIFAR100 | 58.87 | 69.95 | 69.31 | 78.49 | 72.37 |
| FMNIST | 85.97 | 87.08 | 89.49 | 93.77 | 89.62 |

scope. Accordingly, we have included extensive baseline comparisons with other recent FL and particularly personalized FL (pFL) methodologies. A comparative performance analysis on popular datasets like CIFAR100 and FMNIST is presented below:

We utilized the public code and adopted the optimal hyper-parameters from FedCR Zhang et al. (2023), subsequently re-running and documenting all baseline performances under the 'noniid' setting. Our proposed Scafflix was reported with a communication probability of $p = 0.3$ and spanned $500$ communication rounds. We set the personalization factor $\alpha$ at 0.3. Based on the results, when focusing solely on the generalization (testing) performance of the final epoch, our method is on par with state-of-the-art approaches such as FedPAC Xu et al. (2023) and FedCR Zhang et al. (2023). However, our primary emphasis lies in demonstrating accelerated convergence.

