# OpenReview forum: "Explicit Personalization and Local Training: Double Communication Acceleration in Federated Learning"
_ICLR.cc/2024/Conference — Submitted to ICLR 2024_

### Official Review · Reviewer_Av21 · 2023-11-01

**Soundness:** 2 fair
**Presentation:** 2 fair
**Contribution:** 2 fair
**Rating:** 3
**Confidence:** 4

**Summary:**

This paper proposes Scafflix, a new algorithm for personalized federated learning (FL) that achieves double communication acceleration. The paper introduces a novel approach to combine explicit personalization with local training (LT), which are two techniques known to accelerate communication in FL separately. It formulates personalized FL as optimizing an objective function that interpolates between local and global models, with a personalization weight α_i for each client i. It designs Scafflix by applying and adapting a recent LT algorithm called Scaffnew to this personalized FL formulation. Experiments on convex logistic regression and nonconvex neural network training demonstrate that Scafflix converges faster than previous personalized FL algorithms like FLIX and federated averaging.

**Strengths:**

This paper introduces the Scafflix algorithm that achieves doubly accelerated communication for personalized federated learning by combining explicit personalization and local training. A major contribution is combining the ideas of personalization with variance reduction techniques in FL methods. The proposed method is general, allowing heterogeneity across clients, and outperforms prior algorithms as demonstrated through experiments on both convex and nonconvex tasks.

**Weaknesses:**

The paper's novelty appears constrained, as it primarily fuses two existing approaches: variance reduction techniques with accelerated gradient descent methods like FedProx[A] or SCAFFOLD[B], and personalization techniques that interpolate between global and local optima, as observed in various algorithms. The advancements made seem incremental rather than groundbreaking. The theoretical contributions remain unclear, especially when distinguishing the challenges and results of the proposed algorithms from predecessors such as FLIX and Scaffnew.

Experimentally, the analysis is restricted to only FedAvg and FLIX. There's a notable absence of comparisons with state-of-the-art (SOTA) acceleration methods such as SCAFFOLD, FedProx, and others. Additionally, SOTA personalization methods like pFedMe[C], PerFedAvg[D], and APFL[E] are left unexplored. This lack of comprehensive benchmarking, both theoretically and experimentally, hampers a complete evaluation of the paper.

[A] Li, Tian, et al. "Federated optimization in heterogeneous networks." Proceedings of Machine learning and systems 2 (2020): 429-450.

[B] Karimireddy, Sai Praneeth, et al. "Scaffold: Stochastic controlled averaging for federated learning." International conference on machine learning. PMLR, 2020.

[C] T Dinh, Canh, Nguyen Tran, and Josh Nguyen. "Personalized federated learning with Moreau envelopes." Advances in Neural Information Processing Systems 33 (2020): 21394-21405.

[D] Fallah, Alireza, Aryan Mokhtari, and Asuman Ozdaglar. "Personalized federated learning: A meta-learning approach." arXiv preprint
arXiv:2002.07948 (2020).

[E] Deng, Yuyang, Mohammad Mahdi Kamani, and Mehrdad Mahdavi. "Adaptive personalized federated learning." arXiv preprint arXiv:2003.13461 (2020).

**Questions:**

In addition to the mentioned concerns, I have the following question:
How do the proposed approaches behave in different levels of heterogeneity? How should we set the personalization parameter? In real use cases in FL settings, doing Hyperparameter optimization similar to the one being done in the paper is not feasible, hence, it is not clear how the method could be useful for different scenarios where we do not know the level of heterogeneity among clients.

---

### Official Review · Reviewer_zbJj · 2023-11-01

**Soundness:** 3 good
**Presentation:** 3 good
**Contribution:** 3 good
**Rating:** 6
**Confidence:** 3

**Summary:**

This paper introduced a new algorithm,  Scafflix, to solve the FLIX formulation for personalized FL. The authors provided extensive theoretical analysis of their proposed method. Experimental evidence were provided to illustrate faster convergence of Scafflix than existing method for solving FLIX. Scafflix also shows better generalization than the original FLIX method and FedAvg.

**Strengths:**

1. The formulation for personalization and solution method in the paper is novel.
2. The paper is well structured. The motivation and high level ideas are well-written.

**Weaknesses:**

1. It would benefit readers to consume the theoretical analysis framework if the authors provide some intuitions into several important quantities used in the paper:
      i. the control variate variables h_i -> is it trying to control the deviate from the local optimal, or other quantities?
      ii. I did not find a reference of how to set the initial control variate variables h^0_i  in the experiment section. Given them they also have the sum-to-zero constraint and they are all vectors, I don't know how to initialize them in practice. Can you provide some insight?
     iii. Why is the communication probability p required?

2. No comparison of FLIX formulation under Scafflix for other personalization formulation.

**Questions:**

Can you clarify on how to set the initial control variate vectors h^0_i. ?

---

### Official Review · Reviewer_65jh · 2023-11-01

**Soundness:** 3 good
**Presentation:** 2 fair
**Contribution:** 3 good
**Rating:** 3
**Confidence:** 3

**Summary:**

The authors propose and discuss two algorithms for FL: Scafflix. They arrive at their algorithm by extending Scaffnew to the setting in which clients can have individual learning rates $\gamma_i$. This individualized Scaffnew termed 'i-Scaffnew' is then applied to the FLIX model of personalized FL, resulting in the 'Scafflix' algorithm. The authors present a convergence analysis of Scafflix in the strongly convex case, as well as an empirical analysis in the convex case as well as NNs on some common FL benchmarks.

**Strengths:**

The paper is well motivated and describes its contributions very explicitly. My background in convergence proofs for FL is somewhat limited, so I will not comment to detailed on it and set my confidence accordingly. The placing into context of this theoretical contribution within the RW section seems well done.
I appreciate the inclusion of a strongly-convex empirical analysis as an attempt to showcase the exact setting that the theory can support - as well as the inclusion of experiments with NNs.

**Weaknesses:**

Focussing on the empirical analyses, I believe there is not enough information to draw conclusions based on these experiments. As I detail below, many open questions remain.
Specifically, I am missing a more detailed discussion around the elements that make Scafflix different from prior work. I.e. what specifically is the "tuning" of i-Scaffnew for FLIX, as well as how do you perform the "individualization" for Scaffnew through $\gamma_i$ in experiments?

**Questions:**

- It seems that Figure 1 experiments on iid data. Under this assumption, it seems no surprise to me that local training (p<1) is beneficial and smaller alpha leads to smaller objective gap, as $x_i^*$ is close to the global optimum $x^*$. I would need to see the case of $\alpha=0$ to see what the benefit of communicating between client is in the first place. Instead, I would like to see experiments with different levels of non-iid-ness; and how the choice of alpha influences convergence to the optimum.
- For the logreg experiments, I don't see the differences between Scafflix and standard FedAvg applied to the FLIX objective. The claim "Scafflix is much faster than GD, thanks to its local training mechanism" seems to be equivalent to saying "p<1 for FedAvg is faster...", which again is no surprise given the iid-ness of the problem. I seem to be missing the point here. The client-specific learning rates, which supposedly is the key theoretical advantage of Scafflix, is not specified and I assume it is identical across clients (which would make sense given the iid assumption with equal amounts of data).
- In case I did miss a key point of the Scafflix component for the logreg experiments, and there is indeed a difference, could you please include results of FedAvg applied to the FLIX objective as a baseline, i.e. including one acceleration component as a baseline? I suggest concentrating on a single $\alpha$ setting as the trend across all $\alpha$ is identical.
- Similarly, the sentences "In accordance with the methodology outlined in FedJax (Ro et al., 2021), we distribute these samples randomly across 3,400 devices." and " The Shakespeare dataset, used for next character prediction tasks, contains a total of 16,068 samples, which we distribute randomly across 1,129 devices." seem to suggest that you are "randomly distributing" data, leading to iid-splits across clients. Scanning the code (I am not familiar with Jax, or FedJax in particular), it seems that the dataset is the "original", meaning non-iid by writer-id (Femnist) or character (Shakespeare). Please clarify the 'random distributing'.
- For the "Baselines" section of 4.2, I am not sure I understand the selection of the learning rates. Do you independently optimize for the local learning rate per-client to achieve highest validation score, or do you fix the learning rate across clients to be the same for finding the $x_i^*$?. More critically, how do you select the $\gamma_i$ for the actual Scafflix algorithm? Since you fix $alpha_i$ to be identical across clients, I expect the $\gamma_i$ to be different across clients - otherwise it would appear that you are running FedAvg with the FLIX objective. What am I missing here?
- Can you comment on the quality of the $x_i^*$ you find? Anecdotally and from my own experience, these models overfit easily due to the small amount of training-data per client.
- In Figure 1, I interpret the objective gap computed with $f(x^k)$ as using the server-side model evaluated on the entire global training-dataset; could you confirm that? For the NN experiments in Figures 2,3 and the appendix, do you equally evaluate the server-side model, here on the concatenation of all clients' test-datasets? While this is certainly interesting and relevant as an indication for test-client performance, I believe a paper about model personalization in FL should compare to the average client-specific models' performances. Specifically, what is the quality of $x_i^*$ (i.e. no communication, no across-clients knowledge sharing) as well as the performance of $\tilde{x}_i^*$, (i.e. the performance of the client-specific personalized model following the FLIX objection as you detailed in the introduction).
- Scafflix as described in Algorithm 1, as well as your theoretical analysis, does not consider client subsampling. For the theoretical analysis, this should be mentioned as a draw-back IMO. For your empirical analysis, please comment on how introducing client subsampling (are you equating this to "batch-size" in 4.4.2?) provides insights about your algorithm Scafflix.
- For Figure 2, does FLIX, which "uses the SGD method" do multiple local updates (i.e. p<1)? If yes, then the difference to Scafflix would to different local learning rates - is that correct?
- Assuming I understood correctly and different local learning rates is a key component of Scafflix, what is the distribution of local learning rates that you find empirically? What is the empirical difference compared to using the same local learning rate (as I assume corresponds to the FLIX baseline?)

---

### Meta-Review · Area_Chair_yUzs · 2023-12-10

**Metareview:**

The paper combines ideas from personalization and variance reduction in FL. This is an interesting, if somewhat incremental contribution. The reviewers raised several concerns regarding the evidence provided in the experiments and how they were set up. A big issue was also the lack of comparison to existing SoTA algorithms. The authors did not submit a response, so these concerns remain unaddressed.

**Justification For Why Not Higher Score:**

Reviewers point to several deficits. There was not response by the authors.

**Justification For Why Not Lower Score:**

N/A

---

### Decision · Program_Chairs · 2024-01-16

Reject